# Acculturation orientations affect the evolution of a multicultural society

E. Yagmur Erten [1,2], Pieter van den Berg [1,3] & Franz J. Weissing [1,4]

The migration of people between different cultures has affected cultural change throughout history. To understand this process, cross-cultural psychologists have used the 'acculturation' framework, classifying 'acculturation orientations' along two dimensions: the willingness to interact with culturally different individuals, and the inclination to retain the own cultural identity ('cultural conservatism'). Here, using a cultural evolution approach, we construct a dynamically explicit model of acculturation. We show that the evolution of a multicultural society, where immigrant and resident culture stably coexist, is more likely if individuals readily engage in cross-cultural interactions, and if resident individuals are more culturally conservative than immigrants. This result holds if some cultural traits pay off better than others, and individuals use social learning to adopt more advantageous cultural traits. Our study demonstrates that formal dynamic models can help us understand how individual orientations towards immigration eventually determine the population-level distribution of cultural traits.

[1] Theoretical Research in Evolutionary Life Sciences, Groningen Institute for Evolutionary Life Sciences, University of Groningen, PO Box 11103 , 9700 CC Groningen, The Netherlands. [2] Department of Evolutionary Biology and Environmental Studies, University of Zurich, Winterthurerstrasse 190, 8057 Zurich, Switzerland. [3] Laboratory of Socioecology and Social Evolution, Department of Biology, KU Leuven, Naamsestraat 59—bus 2466, 3000 Leuven, Belgium. [4] Netherlands Institute for Advanced Study in the Humanities and Social Science (NIAS-KNAW), PO Box 108551001 EW Amsterdam, The Netherlands. E. Yagmur Erten and Pieter van den Berg contributed equally to this work.  Correspondence and requests for materials should be addressed to F.J.W. (email: F.J.Weissing@rug.nl)

Migration between human populations can affect the cultural repertoire of both immigrant and resident (host) groups in a number of ways. In some cases, immigrants adopt cultural variants that are present in the resident population. For example, the diet of Mexican immigrants in the United States is more similar to the diet of non-Hispanic white Americans than it is to the diet of Mexicans[1]. Immigrant groups can also influence the resident culture. For example, the immigration of approximately 100,000 Chinese immigrants into nineteenth-century Peru substantially influenced Peruvian cuisine. This is evidenced by the culinary tradition of 'chifa', which is rooted in Chinese cuisine but incorporates Peruvian elements, and by the general importance of rice as one of the main carbohydrate staples in contemporary Peru[2]. Other outcomes are also possible: immigrant and resident culture may mix to produce new cultural variants such as Creole languages, or they may retain much of the original culture and often hardly change over long periods of time, as in ethnic enclaves such as Klein-Ankara in Berlin or Chinatowns in the United States and United Kingdom[3].

In the social sciences, cultural changes that take place as a result of migration have traditionally been studied in the framework of 'acculturation'. Acculturation is defined as 'those phenomena which result when groups of individuals having different cultures come into continuous first-hand contact, with subsequent changes in the original cultural patterns of either or both groups'[4]. Berry[5] distinguishes four different acculturation orientations, as resulting from two tendencies: the tendency to maintain the own cultural identity and the tendency to establish relations with the other cultural groups (Fig. 1). Henceforth, we will refer to these dimensions as 'degree of cultural conservatism' and 'interaction tendency', respectively. To be clear, we use the term 'cultural conservatism' in a strict sense, as the tendency to conserve one's culture (so not in the broader political sense that is associated with the term 'conservatism'). According to Berry[5], a high degree of cultural conservatism can either result in an integration orientation (in case of high interaction tendency) or in a separation orientation (in case of low interaction tendency), while a low degree of cultural conservatism can result in an assimilation orientation (in case of high interaction tendency) or a marginalisation orientation (in case of low interaction tendency).

Although Berry's classification is useful, its societal implications are not immediately obvious, for at least two reasons. First, acculturation orientations may differ between individuals in both immigrant and resident groups, and may change over time[3,5,6]. It is far from obvious how the acculturation process might unfold in a population that is mixed and/or dynamic with respect to acculturation orientations. Second, there may be mismatches in the orientations of the immigrant and resident groups; for example, immigrants may prefer to integrate (maintaining their cultural identity), whereas residents may prefer immigrants to assimilate (adopting the resident culture[3]). The societal outcome of the acculturation process in case of such mismatched acculturation orientations is not straightforward to ascertain. For example, it is not clear under which circumstances a multicultural society, in which both immigrant and resident traits stably coexist, is likely to emerge.

Given the dynamic and complex nature of the acculturation process, it is likely that verbal reasoning alone is insufficient to fully appreciate the societal implications of migration on cultural change. The construction of a formal model requires the identification of the important parts of the system and the relationships between them, and forces the scholar to explicitly consider the assumptions that have to be made when developing theory. This facilitates the development of a sharper intuition about the system, and helps avoid mistakes that are easily made when reasoning verbally about complex dynamic systems (see ref. [7] about the benefits of using formal models in the behavioural sciences). Hence, a theoretical modelling approach is needed to aid our understanding. In this study, we develop a model to investigate the dynamics of cultural change that result from migration, depending on the acculturation orientations that are present in the society. To do this, we make use of ideas and techniques from the field of cultural evolution, where there is a significant tradition of studying cultural change using formal methods.

The study of cultural evolution is based on the idea that cultural change is, to a certain extent, a process analogous to genetic evolution[8–12]. Cultural change can be considered as a type of adaptive evolution, because it meets the three basic requirements of evolution by natural selection[13]: variation (cultural traits differ between individuals), differential persistence (some cultural traits may spread more readily than others) and inheritance (cultural traits are transmitted between individuals through forms of social learning). There have been some investigations into the role of migration in the process of cultural evolution[9,14–16], but the effect of acculturation orientations on cultural change has not yet been investigated within this framework.

As social learning is the mode of cultural transmission, the ways in which individuals learn from each other have received much attention in the field of cultural evolution[17]. It has been shown that individuals use various social learning strategies, including 'frequency-based learning', where the probability of acquiring a cultural trait depends on the frequency of that trait in the population (e.g., conformism), and 'success-based learning' where individuals preferentially adopt traits from successful individuals[18,19]. The social learning strategies people use may differ between individuals[19] and between cultures[20], and these differences can have substantial ramifications for the outcome of social interactions[21]. To come to a full appreciation of how cultural traits may change in a population with migration, social learning strategies must be part of the equation.

We present a simple model to systematically assess how acculturation orientations and social learning strategies affect cultural change in populations with continuous immigration. We consider a resident population with a constant influx of immigrants, and assume that these two types originally differ in

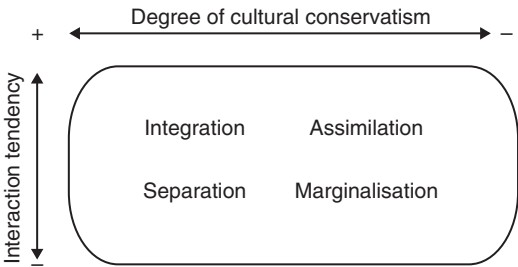

**Fig. 1** Berry's classification of acculturation orientations[5]. The four acculturation orientations result from the two dimensions shown on the axes: how much importance individuals give to maintaining their own cultural identity ('degree of cultural conservatism', horizontal axis) and how much importance they give to establishing interactions with other cultures ('interaction tendency', vertical axis). The four individual orientations can be interpreted as corresponding to possible outcomes of the acculturation process: immigrants are either participating members of the host society that have kept their cultural identity (integration), participating members of the host society that have adopted the host culture (assimilation), socially segregated members of society that have kept their cultural identity (separation), or socially segregated members of society that have not kept their own culture (marginalisation)

the cultural traits they express. Over time, individuals' cultural traits may change through interactions with others in the population, depending on their acculturation orientations. Their interaction tendency determines how likely they are to interact with individuals from the other cultural group, and their degree of cultural conservatism determines how likely they are to change their cultural trait as a result of such interactions. We investigate how the outcome of the model depends on model parameters such as migration rate and the acculturation orientations associated with the resident and immigrant cultures.

We consider three versions of the model, which differ in the assumptions they make on social learning. We start with a simple baseline model, in which we consider two cultural traits (initially associated with residents and immigrants, respectively), that are in principle completely arbitrary (i.e., they do not confer any advantages or disadvantages to the individuals that express them). For example, one might think of the different ways the same ingredient (e.g., potato) is used in the cuisines of various cultures (e.g., *poutine* in Canada, *rösti* in Switzerland or *stamppot* in the Netherlands). In this version of the model, the probability that individuals change their cultural traits is fully dependent on their acculturation orientation (social learning strategies are absent in this case). In the second version of the model, we consider cultural traits that have some kind of functional relevance, considering cases where either the immigrant or the resident cultural trait is more advantageous than the other (i.e., confers a payoff advantage to the individual expressing it, such as the usage of gunpowder weapons in warfare). With this model, we investigate how success-based learning strategies affect the spread of cultural traits. In the third version of the model, we study cases where the payoffs of cultural traits are not constant, but depend on the population constitution. Here, we consider both the case where the most common trait has a payoff advantage (coordination, e.g., driving on the left or right side of the road, or salutations with a handshake vs. a bow), and the case where the least common trait is superior (complementation, e.g., practicing a rare trade and therefore having less competition). Throughout, we track the frequencies of the two cultural traits, and determine whether cultural evolution leads to the fixation of either of the traits, or maintains a multicultural society in which both cultural traits coexist.

We find that both the interaction tendency and the degree of cultural conservatism considerably affect the distribution of resident and immigrant culture, across all scenarios we investigated. Specifically, we find that the stable coexistence of both immigrant and resident culture in a single population (i.e., a multicultural society) is more likely if individuals of both cultural types are relatively willing to interact with each other, and if resident culture is associated with stronger cultural conservatism than immigrant culture. These general patterns hold both in the presence and in the absence of payoff consequences associated with both types of cultural traits. These results show that, at least under the assumptions of our model, acculturation orientations have a robust effect on the stability of multiculturalism.

## Results

**The model**. We consider a finite population of $N$ individuals, which initially consists entirely of residents (see Table 1 for an overview of all model parameters). There is a constant influx of immigrants, and residents and immigrants are initially distinguished by an observable cultural trait that can be subject to change. We refer to individuals that carry the cultural trait initially associated with the residents as 'type R', and individuals that carry the cultural trait initially associated with the immigrants as 'type I' (all immigrants initially carry the same cultural trait). An individual's cultural trait may change through interactions with other individuals. Whether this occurs depends on the acculturation orientation of the individual, which is determined by the two orientational dimensions: the degree of cultural conservatism and the interaction tendency (see Fig. 1).

The model is event based; in each time step, one individual (the 'focal individual') is drawn at random from the population (with probability $\frac{1}{N}$). Next, the focal individual is either replaced by an immigrant (an 'immigration event', occurring with probability $m$), or else paired with a random other individual from the population (an 'interaction event', occurring with probability $1-m$). Hence, $m$ gives a measure of the probability of a migration event relative to an interaction event (rather than denoting the probability of a migration event happening per unit time; see Supplementary Note 1 for a generalised model where the probabilities of migration and interaction are modelled as separate rates, leading to very similar results as the original model). In case of an interaction event, it is not necessarily the case that an interaction actually occurs—this depends on the interaction probability $X$ (which is a function of the interaction tendencies of both individuals). If two individuals of different cultural types actually interact, the focal individual changes its cultural type with probability $S$ (which is a function of its cultural conservatism and social learning strategy). For detailed specifications of the interactions probabilities and the probabilities of changing cultural type, see the 'Methods' section. We implemented this model in two ways: (1) as a deterministic model, where the change in type frequency is described by the following

| Symbol | Meaning |
|---|---|
| $N$ | Population size |
| $m$ | Migration rate (probability of migration event in each time step; interaction event occurs with complementary probability $1-m$) |
| R | Cultural trait initially associated with the resident |
| I | Cultural trait initially associated with the immigrant |
| $p_i$, $p_r$ | Relative frequency of type I and R, respectively |
| $x_{ii}$, $x_{rr}$ | Interaction tendency with the same type (of I and R, respectively) |
| $x_{ir}$, $x_{ri}$ | Interaction tendency with the other type (of I and R, respectively) |
| $X_{ii}$, $X_{rr}$, $X_{ir}$, $X_{ri}$ | Interaction probability of two individuals |
| $c_i$, $c_r$ | Degree of cultural conservatism of type I and R |
| $\Delta c = c_r - c_i$ | Difference between the degree of cultural conservatism of type I and R |
| $S_i$, $S_r$ | Probability to change one's type in an interaction with the other type for type I and type R individuals, respectively |
| $\Delta S = S_r - S_i$ | Difference between the changing probability of type I and R |
| $W_i$, $W_r$ | Transmission advantage ('payoff') of trait I and R, respectively |
| $\Delta W$ | Difference between the payoff of trait I and R |

**Table 1 Model notations**

differential equation:

$$\frac{dp_i}{dt} = \frac{1}{N} \cdot [\underbrace{m \cdot (1 - p_i)}_{\text{immigration}} + \underbrace{(1 - m) \cdot [(1 - p_i) \cdot p_i \cdot X_{ir} \cdot (\Delta S)]}_{\text{interaction}}]$$

(1)

and (2) in an individual-based simulation that takes stochasticity into account. The basic rationale behind equation is as follows. If an immigration event takes place, the focal individual is replaced by an immigrant individual of type I (so there is no change in $p_i$ if the focal individual is of type I). Hence, immigration events increase the frequency of the type I trait ($p_i$) in the population at the rate $\frac{1}{N} \cdot m \cdot (1 - p_i)$. If an interaction event occurs, the cultural trait of the focal individual only changes (with probability $S$) if it is paired with an individual of the other type (which occurs with probability $(1 - p_i) p_i$) and an interaction actually takes place (which happens with probability $X_{ir}$). The net rate of change in $p_i$ resulting from actual interactions is given by $\Delta S$, which is arrived at by subtracting the rate at which type I individuals change to type R (given by $S_i$) from the rate at which type R individual change to type I (given by $S_r$). Hence, the overall rate at which interaction events change $p_i$ is given by $\frac{1}{N}(1 - m) \cdot [(1 - p_i) \cdot p_i \cdot X_{ir} \cdot (\Delta S)]$. For simplicity, we assume that acculturation orientations are linked to cultural traits, so if individuals change their cultural trait, they also change their acculturation orientation (adopting the relevant values for interaction tendencies and cultural conservatism).

To check whether there is general agreement between our analytical model and the simulations, we compare the results of both for the baseline version of the model (in which there are no payoff differences between the cultural traits). For the two versions of the model in which the cultural traits are not equivalent in terms of payoffs, we only show results for our analytical model (see 'Methods' section for a detailed specification of these scenarios and of the social learning strategies that individuals are assumed to use in them).

**No payoff differences**. In the baseline version of the analytical model, when the two cultural traits do not have any consequences

in terms of payoffs, we find two equilibria (equating the right-hand side of equation (1) to zero): one at $p_i^* = 1$ (where the immigrant trait spreads to fixation), and the other at:

$$p_i^* = \frac{m}{1 - m} \cdot \frac{1}{\Delta c} \cdot \frac{1}{X_{ir}},$$

(2)

where both the resident and the immigrant traits coexist in the population. In the latter (internal) equilibrium, $p_i^*$ is positively related to $m$ and negatively related to $\Delta c$ and $X_{ir}$. This makes intuitive sense: a higher migration rate (i.e., a higher probability of migration relative to the probability of interaction) will result in a higher equilibrium frequency of the immigrant cultural trait ($p_i^*$), whereas a relatively high cultural conservatism of type R individuals ($c_r > c_i$, which means $\Delta c > 0$) will result in a lower equilibrium frequency of the immigrant cultural trait. Also, with increased interaction probability between both types ($X_{ir}$), type I individuals (that are initially rare and are therefore mostly paired with type R individuals) are less likely to retain their cultural traits, resulting in a lower equilibrium frequency of the immigrant cultural trait. A graphical representation of equilibrium frequencies is given in Fig. 2 (for both for the analytical model and for the simulations).

The internal equilibrium exists and is stable only if:

$$\frac{m}{1 - m} < \Delta c \cdot X_{ir}.$$

(3)

This condition can only be satisfied if $\Delta c > 0$. In other words, coexistence of both traits is only possible if individuals of type R are more culturally conservative than individuals of type I. For increasing migration rates, stable coexistence of both types requires either a relatively higher cultural conservatism of the resident type (i.e., a larger value of $\Delta c$), or a higher probability that individuals from the different types interact (i.e., a larger value of $X_{ir}$). In line with this, the equilibrium frequency of type R increases with an increase in either of these two quantities.

**Constant payoff differences**. When the cultural traits are associated with payoff differences, we assume individuals will be more likely to adopt the cultural trait of their interaction partner if it is

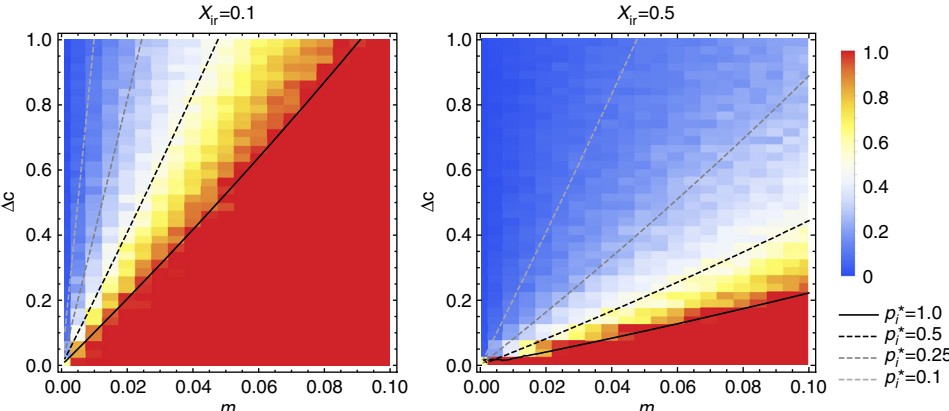

**Fig. 2** Equilibrium frequency of the type I trait in the baseline model. The equilibrium frequency of the trait that is initially associated with immigrants (type I) in the baseline model, in relation to the migration rate ($m$) and the difference in cultural conservatism between both types ($\Delta c = c_r - c_i$). Left panel: low interaction probability between individuals with different cultural traits ($X_{ir} = 0.1$); right panel: relatively high interaction probability ($X_{ir} = 0.5$). Lines depict the equilibrium frequency of the type I trait in the analytical model. For parameter combinations below the solid line ($p_i^* = 1.0$), type R goes extinct. Above the solid line, a stable polymorphism is reached, where the equilibrium frequency of type I is positively related to $m$ and negatively related to $\Delta c$. Note that we only consider migration rates above zero ($m$ ranges between 0.001 and 0.1). Hence, since there is always a constant influx of type I individuals, the type I trait can reach very low values for low values of $m$, but can never go fully extinct for any of the parameter combinations. The heat map indicates the results of individual-based simulations. Each coloured block shows the frequency of the immigrant trait after $10^6$ events, averaged over 10 simulation runs. Simulations and analytical predictions are in good agreement

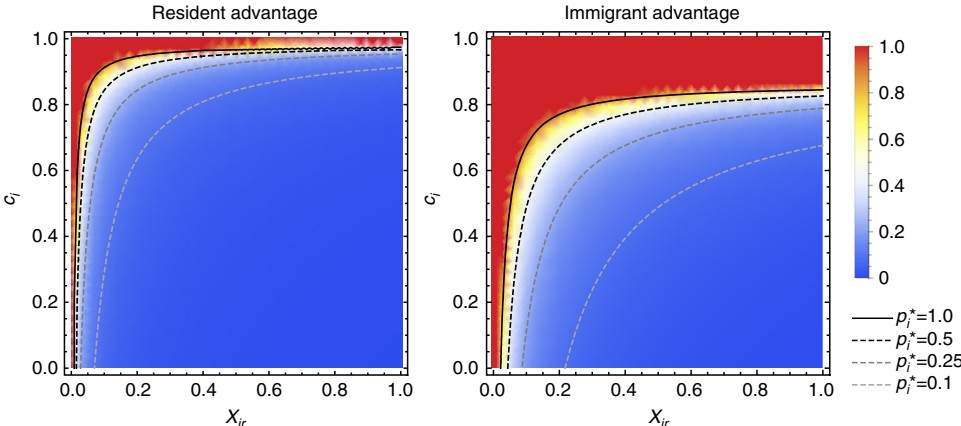

**Fig. 3** Equilibrium with constant payoff differences. The equilibrium frequency of trait that is initially associated with immigrants (type I) when it has a payoff advantage (immigrant advantage, right) or a payoff disadvantage (resident advantage, left). Both lines and colours depict the equilibrium frequencies of the immigrant trait as calculated using the analytical model. Solid lines show the boundary between the coexistence equilibrium and the equilibrium where the type I trait becomes fixed ($p_i^* = 1$). In both graphs $c_r = 0.95$, $W_0 = 1$ and $m = 0.01$

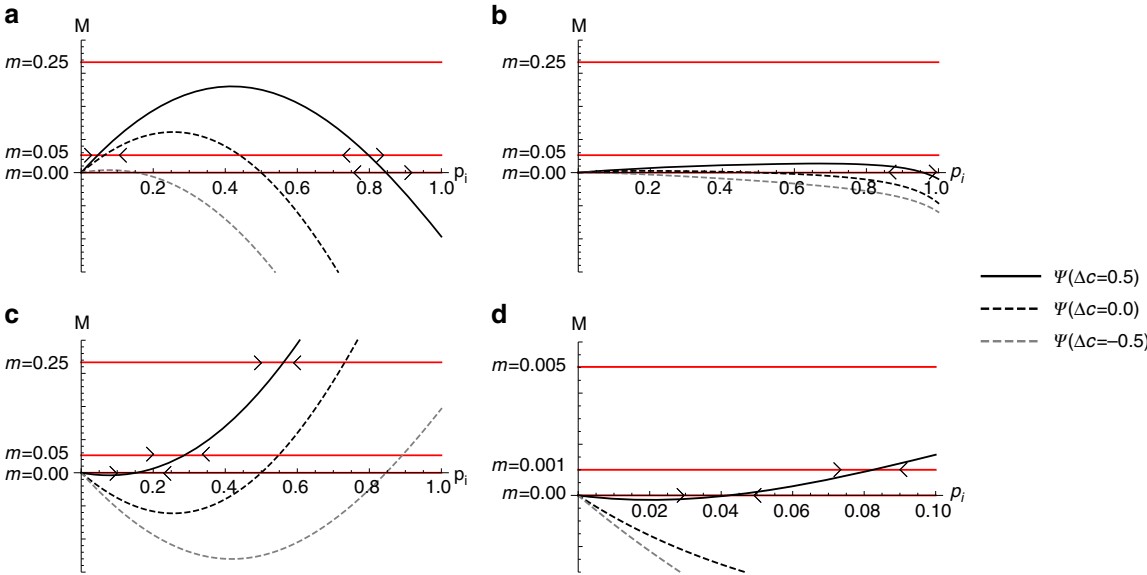

**Fig. 4** Equilibrium structure for the scenarios of coordination and complementation. The payoff of a cultural trait increases with its perceived frequency for the coordination scenario (**a**, **b**) and decreases for the complementation scenario (**c**, **d**). **a**, **c** individuals are equally likely to interact with both types ($X_{ii} = X_{rr} = X_{ir} = 1.0$); **b**, **d** individuals are more likely to interact with individuals of the same type ($X_{ir} = 0.1$; $X_{ii} = X_{rr} = 1.0$). Lines indicate how the left-hand side (red) and the right-hand side (black) of the equilibrium condition $M = \Psi(p_i)$. Equilibria are present when red and black lines cross; arrows depict the stability of the equilibria (shown only for $\Delta c = 0.5$). In both scenarios, coexistence equilibria are more likely to exist if individuals from different types are relatively likely to interact with each other (**a**, **c**), and when the resident type is relatively culturally conservative (solid lines)

associated with a higher payoff than their own cultural trait. Hence, changing probability $S$ is now determined by both cultural conservatism and the payoff difference between both traits (see 'Methods' section for details). In the scenarios of resident advantage and immigrant advantage, the payoff difference ($\Delta W$) between both cultural traits is constant. Consequently, the difference between the changing probabilities of the two types ($\Delta S$) is also constant. In the analytical model, we find two equilibria: one at $p_i^* = 1$, as in the model without payoff consequences, and the other at:

$$p_i^* = -\frac{m}{1-m} \cdot \frac{1}{\Delta S} \cdot \frac{1}{X_{ir}}, \tag{4}$$

where both cultural traits coexist. This equilibrium exists and is

stable only if:

$$\frac{m}{1-m} < -\Delta S \cdot X_{ir}. \tag{5}$$

$\Delta S$ depends both on the payoffs associated with the cultural traits and on the cultural conservatism of the two types.

Figure 3 shows the equilibrium frequencies of the immigrant trait in the analytical model both for the resident advantage scenario (left panel) and the immigrant advantage scenario (right panel). Notice that the parameters are chosen such that the immigrant trait does not easily fixate: the migration rate is low ($m = 0.01$) and the resident trait is associated with a high cultural conservatism ($c_r = 0.95$). Yet, even under these conditions, the immigrant trait can spread to fixation in both the immigrant advantage and resident advantage scenarios if the probability of interaction between both types is very low (i.e., for low $X_{ir}$). In

other words, even if the resident cultural trait is superior, it will be lost from the population if immigrants and residents rarely interact with each other.

**Frequency-dependent payoff differences.** In the scenarios of coordination and complementation, payoffs depend on the perceived frequencies of the resident and immigrant types in the population. This leads to a substantial increase in the complexity of the model, causing the equilibrium analysis of these scenarios to be a tedious exercise. Therefore, we use a numerical approach to identify equilibria. Again, we find the equilibrium in which the immigrant type is fixed ($p_i^* = 1$), and we identify an internal equilibrium at:

$$\frac{m}{1-m} = -p_i^* \cdot X_{ir} \cdot \Delta S. \tag{6}$$

For illustrative purposes (see Fig. 4), we summarise each of the sides of this equation into a single parameter, one of which depends on only the migration rate and denotes the ratio of the probability of a migration event to the probability of an interaction event:

$$M = \frac{m}{1-m} \tag{7}$$

and the other dependent on the other parameters, which include acculturation orientation, social learning strategy and the population constitution:

$$\Psi(p_i) = -p_i^* \cdot X_{ir} \cdot \Delta S, \tag{8}$$

so that an internal equilibrium exists if

$$M = \Psi(p_i). \tag{9}$$

Internal equilibria can be realised for both the coordination and complementation scenarios (Fig. 4). In the coordination scenario, if all interaction probabilities are high ($X_{ii} = X_{rr} = X_{ir} = 1.0$), and the migration rate is sufficiently low, we find two internal equilibria (Fig. 4a), one of which is a stable coexistence equilibrium. For higher migration rates, the internal equilibria disappear and the type I trait spreads to fixation. Furthermore, if the type I trait confers a high cultural conservatism, it has a higher probability to fixate in the population. If the two types are relatively unlikely to interact with each other ($X_{ir} = 0.1$; $X_{ii} = X_{rr} = 1.0$), the type R trait needs to confer a higher cultural conservatism for both traits to coexist in the population, even for low migration rates (Fig. 4b), rendering it relatively likely that the type I trait spreads to fixation in this case.

In the complementation scenario, only a single internal equilibrium can exist (Fig. 4c, d), in which case it is always a stable coexistence equilibrium. Also here, stable coexistence is much more likely to be realised if both types are likely to interact with each other (Fig. 4c), than if they are not (Fig. 4d).

## Discussion

In this study, we combined dynamic modelling tools derived from evolutionary biology within the conceptual framework of acculturation from cross-cultural psychology to study how migration affects cultural change. Adopting Berry's classification of acculturation orientations in a cultural evolution model, we studied how cultural conservatism and the tendency to interact with individuals from other cultures affect cultural change in a population with continuous immigration. In addition, we considered a number of scenarios in which cultural traits have different payoff consequences, and investigated how acculturation

orientations and social learning strategies interact to produce cultural change.

Our model generated a number of insights into how acculturation orientations can affect the establishment of immigrant cultural traits in a resident population. As might be expected, the cultural conservatism of both immigrants and residents (the tendency to retain one's cultural heritage) has a clear impact on the establishment and spread of immigrant culture. Moreover, the rate at which individuals expressing different cultural traits interact plays a crucial role. If immigrants and residents do not tend to interact much, the immigrants will not adopt the resident culture, even if they are in principle willing to change, and/or if the resident culture is associated with better payoffs.

Our model predicts that the emergence of a stable multicultural society, in which both immigrant and resident cultural traits coexist, is dependent on two main factors. First, the probability of stable coexistence increases with the willingness of immigrants and residents to interact with each other. Second, coexistence is more likely when the immigrant cultural trait is associated with a higher willingness to change than the cultural trait of the resident (i.e., if residents are more culturally conservative). Interestingly, these results hold regardless of the payoffs that are associated with the different cultural traits. We predict that constant payoff differences (resident advantage or immigrant advantage) are only of importance if individuals of both cultures interact with a frequency that is sufficiently high (and even then, they are far from all-determining). When the payoffs of cultural traits increase with their frequency (coordination) and in the opposite scenario where payoffs decrease with frequency (complementation), many outcomes are possible, but our overall conclusions regarding the probability of the emergence of a multicultural society hold in both these cases.

Using a cultural evolution approach to study acculturation involves a conceptualisation of culture that diverges from how culture is typically studied in cross-cultural psychology. Berry[22] defines culture as 'a complex set of interrelated independent variables' ('independent' meaning 'having an existence outside any particular individual'). In contrast, within the cultural evolution framework, culture is treated as a set of transmissible characteristics that ultimately reside in individuals. Although acculturation orientation is a concept that originates from cross-cultural psychology, it is relatively straightforward to see how it relates to the spread of culture between individuals. Our model shows that the individual-centred approach from cultural evolution can be used to investigate the importance of such concepts for the process of cultural change. Having said that, our model highly simplifies immigrant and resident culture to only two discrete cultural traits. By doing this, we disregard the fact that cultural traits are often interrelated, and embedded in a larger cultural repertoire—phenomena that typically receive more attention in the tradition of cross-cultural psychology. Hence, although our model makes some first steps towards integrating both perspectives, there is still plenty of scope for progress in this regard.

To be able to come to an insightful interpretation of model outcomes, it is necessary to make simplifying assumptions. In this study, we assumed that the population is well mixed (individuals are equally likely to encounter any other individual in the population), although we did allow for non-random interaction probabilities as a result of individuals' acculturation orientations. This 'mass action' principle is a standard simplifying assumption of many models in biology (e.g., epidemiological models), but it is hardly a realistic reflection of actual interaction structures in human societies. In reality, spatial configuration and social network structure strongly determine interaction structure[23], and are known to have a significant impact on ecological and

evolutionary dynamics[24], including cultural evolution (e.g. refs. [25,26]). In our case, we might expect that spatial structure leads to clustering of residents and migrants if the interaction probabilities between both types are low (as in ethnic enclaves). Thus, spatial structure may allow for the coexistence of both immigrant and resident cultural traits across wider range of conditions. There are models that explore interactions between different cultures in a spatial context, such as Schelling's segregation model[27] and models that look at the propagation of cultural traits in space[28]. However, further work is needed to study how migration may affect cultural change and the distribution of cultural variants in explicitly spatially structured (or network-structured) populations, when cultural change is a result of acculturation and/or social learning.

In this study, we focus on differences between resident and immigrant culture—we do not consider any individual differences in cultural traits or acculturation orientations within cultural groups. As part of this approach, we have also assumed that cultural trait and acculturation orientation are inextricably linked; if individuals change their cultural trait, they also change their acculturation orientation along with it. We have made these choices to keep our model relatively simple and our results easily interpretable, but this does not mean that we think individual differences within cultures are not important to acculturation. Although there are cultural traits that are indeed much more variable between cultures than within them (e.g., using chopsticks vs. western style utensils), within-culture individual differences are certainly relevant with respect to various cultural traits (e.g., the extent to which people are religious). In addition, previous work has shown that the outcome of the (individual) acculturation process is significantly affected by various individual-level variables[29]. Indeed, we think that incorporating individual differences in dynamically explicit models of acculturation would be an interesting avenue of further research.

Our model assumes a fixed population size and a constant influx of a single type of immigrants into the population. It is important to note that these assumptions make it impossible for the immigrant trait to go extinct, even if it is associated with a relatively poor payoff or if immigrants are much less culturally conservative than residents. Indeed, because of the incessant immigration in our model, immigrant culture will always become fixed in the population unless counteracted by mechanisms such as relatively high resident cultural conservatism or complementarity of cultural traits. Especially when the probability that immigrants and residents interact is low, the mechanisms that counteract the spread of immigrant culture can easily be overwhelmed by the spread of immigrant culture caused by the immigration rate alone. In reality, populations may grow or shrink, migration pressures fluctuate over time and the cultural type of immigrants may often change over the course of time. The incorporation of such factors could strongly affect our results. For example, if we would assume a single burst of immigration rather than a constant rate, immigrant culture would equilibrate at lower levels or even go extinct (of course depending on the specific assumptions about payoffs associated with both traits).

Our model assumes that acculturation orientations are fixed, and cannot change over time. In reality, increasing migration rates (and resulting large immigrant communities) can lead to increases in negative attitudes towards migrants among the resident population[30]. If this results in changing acculturation orientations, this can have a significant impact on the dynamics of cultural change. In similar vein, mismatches may occur between immigrant acculturation orientations and resident acculturation expectations (i.e., how residents or the society as a whole would prefer the immigrants to act)[3,31]. The interactive acculturation model[32] predicts that such mismatches can have a significant impact on the process of acculturation and can result in conflicts. Our modelling framework lends itself well for studying the consequences of changing acculturation orientations and mismatching acculturation orientations and expectations. Indeed, allowing for conditionally changing acculturation orientations (e.g., as a function of the frequency of the immigrant trait or the migration rate) provides one of the most promising extensions of our current model. Longitudinal studies of acculturation orientations and expectations would be an ideal way to provide an empirical basis for such models.

## Methods

**Specification of interaction probabilities**. The interaction probability $X$ is determined by the product of the interaction tendencies of both interaction partners. If an individual of type I and an individual of type R are paired in an interaction event, we calculate the interaction probability as:

$$X_{ir} = x_{ir} \cdot x_{ri}. \tag{10}$$

Similarly, the interaction probability between two individuals of the same type is calculated as $X_{ii} = x_{ii} \cdot x_{ii}$ for type I, and $X_{rr} = x_{rr} \cdot x_{rr}$ for type R.

Only interactions between individuals with different cultural traits can result in a change in the cultural trait of the focal individual. Whether this happens depends on the focal individual's changing probability $S$, which is a function of its cultural conservatism and its social learning strategy (see below).

**Specification of changing probabilities**. We allow for the possibility that in case of an interaction, one of the traits is more easily transmitted than the other. To model this, we ascribe 'payoffs' $W_r$ and $W_i$ to the traits and assume that the ease of transmission of a trait is related to the payoff difference between the traits ($\Delta W = W_r - W_i$). To be more specific, an individual will change its cultural trait with changing probability $S$, which is the product of its willingness to change (determined by its cultural conservatism $c$) and its changing bias (determined by the payoff difference $\Delta W$). For type I individuals, this probability is given by:

$$S_i = \underbrace{(1 - c_i)}_{\text{willingness to change}} \cdot \underbrace{\frac{2}{1 + e^{-\Delta W}}}_{\text{changing bias}}. \tag{11}$$

For type R individuals, it is given by:

$$S_r = (1 - c_r) \cdot \frac{2}{1 + e^{\Delta W}}. \tag{12}$$

Following the standard way of modelling success-based learning in cultural evolutionary theory (e.g., ref. [33]), we model the changing bias as an increasing S-shaped function of the difference between the own payoff and the payoff of the other type, with the magnitude of the difference between the payoffs $\Delta W$ determining the slope of the function. We denote the difference between changing probabilities of type R and type I as $\Delta S = S_r - S_i$. The factor 2 in the changing bias ensures that $S$ simply reduces to $1-c$ if both cultural traits have equal payoffs (i.e., if $W_r = W_i$); if this is the case, $\Delta S = c_i - c_r = -\Delta c$.

**Payoff consequences**. We consider four scenarios in which the two cultural traits have payoff consequences, affecting their transmission probabilities to other individuals through success-based learning. In the first two scenarios, 'resident advantage' and 'immigrant advantage', we assume that payoffs $W_r$ and $W_i$ are constant, but one is larger than the other (the superior payoff is set to $W_0 > 0$, and the inferior payoff is set to 0). In the next two scenarios, we assume that the payoff associated with a cultural trait is dependent on its perceived frequency in the population. In the 'coordination' (or conformism) scenario, individuals are more likely to adopt a cultural trait if they perceive its frequency to be higher (i.e., the payoff of a cultural trait increases with its perceived frequency). Conversely, in the 'complementation' scenario, payoff decreases with perceived frequency.

We assume that individuals do not know the exact population constitution, but perceive the frequencies of the cultural traits to be equal to the frequencies with which they encounter them in interactions. We denote perceived frequency as $p_{x|y}$ (the perceived frequency of type X, from the point of view of type Y), and calculate it as:

$$p_{x|y} = \frac{p_x \cdot X_{xy}}{p_x \cdot X_{xy} + (1 - p_x) \cdot X_{yy}}. \tag{13}$$

This is simply the conditional probability that the focal individual interacts with an individual of type X, given that it is of type Y. The payoffs in the coordination and the complementation scenarios are directly proportional to the perceived frequencies: in the coordination scenario, $W_r = W_0 \cdot p_{r|r}$ and $W_i = W_0 \cdot p_{i|i}$; in the complementation scenario, $W_r = W_0 \cdot p_{i|r}$ and $W_i = W_0 \cdot p_{r|i}$.

**Individual-based simulation parameters**. In the individual-based model, migration, interaction and social learning were implemented exactly as described for the analytical model. We use this model to verify if our analytical results for the baseline model still hold in a finite population in which stochasticity plays an important role. We simulated a population of size 1000 individuals, systematically varying the migration rate $m$ (ranging between 0.001 and 0.1 with step size 0.005), parameters $c_i$ and $c_r$ (ranging between 0.001 and 1.0 with step size 0.05), and $X_{ir}$ (for values 0.1 and 0.5). For each of the resulting parameter combinations, we ran 10 replicate simulations of $10^6$ events.

**Code availability**. The simulation code used in this study has been made available at GitHub (https://github.com/yagmurerten/migration2017).

**Data availability**. The mathematical models, data generated by the simulations and the processing script are deposited in figshare with the identifier (https://doi.org/10.6084/m9.figshare.4748269).

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

## Acknowledgements

P.v.d.B. is supported by a Rubicon fellowship (File no. 019.2015.2.310.041) of the Dutch Science Foundation (NWO).

## Author contributions

E.Y.E., P.v.d.B. and F.J.W. designed the study, E.Y.E. implemented the model and carried out the computer simulations, E.Y.E., P.v.d.B. and F.J.W. analysed the model and wrote the paper.

## Additional information

**Competing interests:** The authors declare no competing financial interests.

