## [Peer Review File · Nature Communications]

Reviewers' comments:

Reviewer #1 (Remarks to the Author):

Review of Nature Communications ms. NCOMMS-16-30372

Page 2 – rather than saying that immigrants adopt the practices of the receiving culture (or vice versa), isn't it some of both? Why is this cast as an either/or proposition?

Page 3 – I would disagree that ethnic enclaves are “relatively unchanging.” A Chinese person coming to a Chinatown in the USA or UK would definitely feel out of place.

The concept of “conservatism” has strong political overtones, and I read it as such in the abstract. I would suggest that the authors choose a different term.

Also on page 3 – interactive models of acculturation have laid out what happens when migrants' cultural orientations do not match with those of the receiving society. Also see John Berry's forthcoming chapter in the Oxford Handbook of Acculturation and Health (already available online).

I like the evolutionary and social learning approaches that the authors adopt. However, is that all there is to the story? If cultural traits are subject to natural selection, then how has New York Jewish culture (to name an example) survived for almost 100 years since the mass immigration of European Jews to New York? I am not sure I would label Judaism as a dominant cultural stream.

I have a hard time with the idea that there are some traits associated with “immigrants” and others associated with “residents.” What about individual differences between and among people in both groups?

Is it premature to mathematize interactions between immigrants and non-immigrants? What about second generation immigrants who are born in the country of residence but raised by foreign-born parents? Do they count as immigrants as well? And how do we know what the “payoff” is for a given trait? Let's take Spanish fluency in the US as an example. In some contexts, Spanish language use is an advantage, whereas in other contexts, it may elicit discrimination. In still other contexts, it may elicit both types of reactions. I am not sure whether a mathematical model can capture these nuances.

On page 9, the authors note that “a higher migration rate will result in a higher equilibrium frequency of the immigrant cultural trait.” Is this really true? Defensiveness on the part of receiving society members is not taken into account here! Are Europeans taking on Islam, and are Americans moving toward Spanish, simply because of population increases? This model does not take into account backlashes against population shifts (e.g., the Donald Trump phenomenon). Would this model have predicted Trump's election in the US or Britain's move to leave the EU?

Further, cultural “conservatism” (we need a different term for this) can be expressed in many different ways. The Trump phenomenon is a more defensive form of cultural conservatism that may not necessarily be equated with gentler forms. How does the model account for these variations in “conservatism”?

Interaction between immigrants and receiving-society members may also be a function of residential segregation – the degree to which immigrants live in separate communities (enclaves) than receiving-society people do. How does the model take this population-level phenomenon into account?

In sum, I like where the authors are going, but their model needs to take historical, social-environmental, and population-level determinants into account. I am not sure whether it does so currently. The fact that the authors’ simulations match their predictions does not mean that they would hold up in a real-world context.

Reviewer #2 (Remarks to the Author):

This paper presents a cultural evolution mathematical model of acculturation (psychological and behavioral changes due to migration) as studied in cultural psychology. The main findings are fairly intuitive: traits brought by migrants are more likely to spread if migration is frequent, and residents are less culturally conservative. There are some nice extensions regarding payoff-related traits and social learning strategies (e.g. conformity) that make useful links to the cultural evolution literature. Overall I think the paper is worthy of publication. It provides a novel, formal treatment of a phenomenon (acculturation) that has been discussed only using verbal arguments in the past. The models seem rigorously analysed and formulated. There is great opportunity for formal cultural evolution models to contribute to this area, and this paper will stimulate that effort.

I have a few recommendations and requests for clarification:

1. There could be a lot more effort to engage with the acculturation literature. I fear that acculturation researchers will be left cold by the mix of intuitive findings and simplifications made in the models. In the discussion I would like to see specific recommendations deriving from the models: are the insights from the models consistent with specific empirical patterns? What do the models suggest are important factors that are not currently considered in acculturation research, but which could be measured? In other words, how exactly can these models guide future empirical work?

2. The term ‘multicultural’ needs to be clearly defined. In the present models it seems to denote an internal equilibrium where some individuals have one trait and other individuals have the other trait. I doubt whether this would count as ‘multiculturalism’ in either the acculturation literature or to lay people. There is no blending of traits, or single individuals holding multiple traits of different cultural origin. Using multicultural in this restricted sense

with no definition will lead to confusion.

3. Please give examples, ideally drawing on the acculturation literature, of real-life traits subject to acculturation that might plausibly be (i) payoff-neutral, as in the first model, (ii) payoff-relevant, as in the second model, and (iii) subject to frequency effects, as in the third model. I felt the paper became very abstract at this point, and could have done with greater connection to the real world.

4. The overview of cultural evolution theory in the introduction is good, but I would cite and discuss Cavalli-Sforza & Feldman (1981) who already modelled the consequences of migration for cultural evolution (albeit not within an acculturation context).

5. Line 83: "Given the dynamic and complex nature of the acculturation process, it is likely that verbal reasoning alone is insufficient to fully appreciate the societal implications of migration on cultural change". I think there could be more elaboration of this point for an acculturation audience who fail to see the value of modelling (see also point 1 above). Paul Smaldino has written some excellent articles recently explaining the benefits of models to psychologists.

6. Line 298: "More strikingly, ...[i]f immigrants and residents do not tend to interact much, the immigrants will not adopt the resident culture" - this seems rather obvious, and does not warrant the superlative "striking"

7. Line 351: "these models" - citations needed

8. Please include all model code for the simulations, and ideally scripts for the analytical model analyses, as supplementary information, for others to check and build upon.

Reviewer #4 (Remarks to the Author):

Review of 'The effect of acculturation on the evolution of a multicultural society'

This is a theoretical model examining the conditions under which we might expect to see the emergence of multicultural societies. This brings together perspectives from cultural psychology and theoretical cultural evolution.

Both the question asked and the perspectives taken are interesting and it is a very well written paper in my opinion. However, there appear to be some important assumptions underlying the results that have not been clearly specified or justified (see below). These will need to be addressed before publication.

Major points.

(1) It is unclear why learning events and migration events should be mutually exclusive. The text does not offer an explanation – is there a reason? There is a probability m of

immigration and a probability $(1-m)$ of a learning interaction (subject to other parameters), so when m is high, learning is less likely. This seems like it would artificially strengthen the effects of increased migration – not only is there more of the I trait to start with, but when m is high, it is less likely to be replaced through interaction and learning.

There is another problem with this assumption. Usually, one would assume that as migration increases, the probability of the resident population interacting with a migrant also increases (this is just a sensible by-product of a well-mixed population). In this case this effect is watered down and that assumption will need to be justified and the consequences made clear at the outset of the paper.

(2) The model assumes that immigration is constant. The I trait is constantly flowing into the population and can also be transmitted through learning. However, the resident R trait can only spread through learning because the resident population cannot reproduce and replace immigrants. When interaction is low, this heavily favours the immigrant trait.

This assumption explains the finding that high levels of interaction favour the persistence of both traits and that low levels lead to fixation of the immigrant trait. If both populations could reproduce and replace one another, one might expect the opposite to be true (i.e. that low levels of interaction would maintain pockets of both traits and that high levels of interaction would favour a higher fitness trait, for example.)

The issue with this is shown clearly in Figure 3 where zero interaction on the left of the x-axis leads to fixation of the immigrant trait. It is also exemplified by Lines: 230-231 : ‘...In other words, even if the resident cultural trait is superior, it will be lost from the population if immigrants and residents rarely interact with each other’ and 311-313 : ‘We predict that constant payoff differences (resident advantage or immigrant advantage) are only of importance if individuals of both cultures interact with a frequency that is sufficiently high (and even then, they are far from all-determining)’, which are both highly counter-intuitive statements. With a strong fitness advantage, no trait should die out, all else being equal.

(3) It is not clear why the simulations (e.g. in Figure 2) so frequently show the I trait dying out when (a) migration should be constant and therefore a lower bound on the frequency of I might reasonably be expected to be m and (b) the analytical model shows two equilibria, neither of which is extinction of the I trait? I’m sure there is a reason but could the authors add some text to the discussion of Figure 2 to address this?

Minor Points

Figure 1. This is interesting and helpful for those not familiar with this theory. It would also be very helpful for the authors to clearly show what each of the four endpoints would mean in their model. It looks like some of them are not possible in the model and it is important to know that not all outcomes are possible. For example, multiculturalism in the model means that both traits survive in some proportion. Both traits cannot exist in one individual so perhaps ‘assimilation’ is not possible but ‘separation’ is possible? This should be made

clear in the figure.

Equation 1. In general, I usually maintain that equations should not appear in the text if not fully explained. Perhaps either point to the methods and omit the equation at this point or explain its terms fully. It does not fulfill a useful purpose at that point in the text.

Figure 2. When m is zero, there is no immigration. If the population is initially all of type R, all boxes in the left hand corner of the plots should be dark blue (a frequency of 0 for type I) – why are they showing a frequency of ~ 0.5 ?

'The effect of acculturation on the evolution of a multicultural society'
Manuscript by: Erten, Van den Berg and Weissing

Response to reviewers

This document contains a point-by-point description of how we revised our manuscript in response to the reviewers' concerns. We are grateful for the reviewers' comments; all three review reports were constructive and have made a significant contribution to improving the manuscript. Reviewer remarks will be printed in black, and a concise description of our response will be printed in blue.

Reviewer #1

Page 2 – rather than saying that immigrants adopt the practices of the receiving culture (or vice versa), isn't it some of both? Why is this cast as an either/or proposition?

We agree that it is important not to set these examples as an either/or situation, and therefore have removed "conversely" from the sentence (line 42). We now give these as two parallel but not necessarily opposite examples. As we then continue by saying there are other possible outcomes and give a few examples including creolization, we assume it is clear that both of the cultures can adopt some traits from each other.

Page 3 – I would disagree that ethnic enclaves are "relatively unchanging." A Chinese person coming to a Chinatown in the USA or UK would definitely feel out of place.

We agree that these ethnic enclaves will differ from the culture of origin in some respects (through the influence of resident culture), although they also still retain many of their original cultural traits. For instance, in Chinatown, one would be served food with chopsticks, although it is true that one would be likelier than in China to be offered western type utensils as well. We have now worded this less strongly; we replaced "relatively unchanging" with "retain much of the original culture and often hardly change over long periods of time" (line 49).

The concept of "conservatism" has strong political overtones, and I read it as such in the abstract. I would suggest that the authors choose a different term.

We recognize this concern, and have considered options for renaming this parameter. In the end, we have chosen to more explicitly explain our use of the term rather than replacing it. In addition, we now avoid confusion by no longer using 'conservatism' as a standalone term, but always using 'cultural conservatism' throughout the text. The reason why we have not replaced the term is that it succinctly conveys the meaning of this variable in our model: the tendency to retain (conserve) one's culture, and we would like to avoid having to resort to more elaborate terminology. We have clarified that we use the term only within the cultural context, in two places. First, we have added a clarification in the abstract (lines 26-27), and second, we now explicitly explain our use of the term in the introduction by adding: "To be clear, we use the term 'cultural conservatism' in a strict sense, as the tendency to conserve one's culture (so not in the broader political sense that is associated with the term 'conservatism')." (lines 60-63).

Also on page 3 – interactive models of acculturation have laid out what happens when migrants' cultural orientations do not match with those of the receiving society. Also see John Berry's forthcoming chapter in the Oxford Handbook of Acculturation and Health (already available online).

We thank the reviewer for directing us to this literature. We added references to the Interactive Acculturation Model by Bourhis et al. (line 420) and to John Berry's chapter in the

Oxford Handbook of Acculturation and Health (line 419) in the last paragraph of our Discussion.

I like the evolutionary and social learning approaches that the authors adopt. However, is that all there is to the story? If cultural traits are subject to natural selection, then how has New York Jewish culture (to name an example) survived for almost 100 years since the mass immigration of European Jews to New York? I am not sure I would label Judaism as a dominant cultural stream.

We fully agree that the cultural evolution approach that we adopt is by itself not sufficient to understand the complex dynamics of cultural change in real societies. Rather, our study provides some general insights about how the (already relatively complex) interaction of acculturation and social learning processes affect cultural change, and under which conditions we should expect a (stably) multicultural society to be a likely outcome. In response to this point, we now place more emphasis on the scope and limitations of our model with a more extensive discussion of factors that are of importance to the process of cultural change, but were not included in our model (last three paragraphs of the Discussion section, lines 375-428).

I have a hard time with the idea that there are some traits associated with “immigrants” and others associated with “residents.” What about individual differences between and among people in both groups?

Our model is individual-based, and individuals from the resident culture can adopt traits initially associated with the immigrants and *vice versa*, so within-group individual differences can emerge over time in our model. However, as the reviewer notes, we do assume that all immigrants and all residents are initially uniform with regard to their cultural traits. The main reason we have made this assumption is to keep our model relatively simple, so that the results are not too difficult to interpret. However, we do agree that individual differences in culture and in acculturation orientations are often relevant, and can significantly impact cultural change. Therefore, we have added a paragraph in the Discussion section to reflect on this issue (lines 395-407).

Is it premature to mathematize interactions between immigrants and non-immigrants? What about second generation immigrants who are born in the country of residence but raised by foreign-born parents? Do they count as immigrants as well?

Our model is cultural and therefore makes no explicit assumptions about biological generations. This could be an interesting addition to the model, but it is beyond the scope of what we are looking at in the current model.

And how do we know what the “payoff” is for a given trait? Let’s take Spanish fluency in the US as an example. In some contexts, Spanish language use is an advantage, whereas in other contexts, it may elicit discrimination. In still other contexts, it may elicit both types of reactions. I am not sure whether a mathematical model can capture these nuances.

We agree with the reviewer that our assumptions regarding how payoffs result from interactions are relatively abstract, and disregard a number of factors that are of influence in real-world situations. Having said that, we do consider two situations where the payoff of a cultural trait depends on the context: coordination and complementation. In the coordination scenario, the payoff of a trait is higher if it has a higher frequency in the population. To follow the language example, this could be the case for speaking the majority language in a country: the more people speak the same language, the higher the benefits of speaking that language (because individuals speaking that language are able to interact with more people). In the complementation scenario, the payoff of a trait is higher if it is rarer in the population. This could be the case for speaking a second language: if fewer people speak this specific second language, individuals that do speak it may have an advantage on the labour market. It is also possible to model more complex payoff consequences of cultural

traits, like in the example the reviewer gives. Our model could relatively easily be adopted to study such cases, but they are beyond the scope of the current study.

On page 9, the authors note that “a higher migration rate will result in a higher equilibrium frequency of the immigrant cultural trait.” Is this really true? Defensiveness on the part of receiving society members is not taken into account here! Are Europeans taking on Islam, and are Americans moving toward Spanish, simply because of population increases? This model does not take into account backlashes against population shifts (e.g., the Donald Trump phenomenon). Would this model have predicted Trump’s election in the US or Britain’s move to leave the EU?

We agree with the reviewer that a negative attitude towards immigrants caused by a high migration rate could have a significant impact on cultural change. In particular, it would be interesting to account for a change and potential mismatch in the acculturation orientations due to this negative attitude. We added a new paragraph to the Discussion section in which we address this issue (lines 413-426).

Further, cultural “conservatism” (we need a different term for this) can be expressed in many different ways. The Trump phenomenon is a more defensive form of cultural conservatism that may not necessarily be equated with gentler forms. How does the model account for these variations in “conservatism”?

As explained previously, we consider “conservatism” only in a relatively strict definition, as the tendency to retain one’s cultural traits (rather than the broader political interpretation). As explained under the third point of this reviewer, we have made this more explicit in both the abstract and the introduction.

Interaction between immigrants and receiving-society members may also be a function of residential segregation – the degree to which immigrants live in separate communities (enclaves) than receiving-society people do. How does the model take this population-level phenomenon into account?

As we discuss in lines 375-394, our model does not take spatial configuration into account explicitly, and can therefore not account for residential segregation (like for instance in Schelling’s segregation model). However, the interaction tendency accounts for this in an implicit way; the lower the interaction tendency of residents and immigrants with each other, more segregated they are in terms of interactions. Indeed, this interactive segregation could in principle be interpreted as a segregation in space.

In sum, I like where the authors are going, but their model needs to take historical, social-environmental, and population-level determinants into account. I am not sure whether it does so currently. The fact that the authors’ simulations match their predictions does not mean that they would hold up in a real-world context.

The aim of our model is not so much to generate predictions as it is to provide insight into how acculturation tendencies and social learning might interact in a dynamic system to produce patterns regarding the distribution of cultural traits. Hence, our conclusions are qualitative (higher resident conservatism and/or interaction tendencies are more likely to maintain both resident and immigrant culture in a single population) rather than quantitative (a prediction of the percentages of resident and migrant culture that should be expected based on specific values for resident and migrant acculturation attitudes). As mentioned to this reviewer in the previous responses, we address this general concern with a more extensive discussion of the scope and limitations of our model (last three paragraphs of the Discussion section, lines 375-428).

Reviewer #2

This paper presents a cultural evolution mathematical model of acculturation (psychological and behavioral changes due to migration) as studied in cultural psychology. The main

findings are fairly intuitive: traits brought by migrants are more likely to spread if migration is frequent, and residents are less culturally conservative. There are some nice extensions regarding payoff-related traits and social learning strategies (e.g. conformity) that make useful links to the cultural evolution literature. Overall I think the paper is worthy of publication. It provides a novel, formal treatment of a phenomenon (acculturation) that has been discussed only using verbal arguments in the past. The models seem rigorously analysed and formulated. There is great opportunity for formal cultural evolution models to contribute to this area, and this paper will stimulate that effort.

I have a few recommendations and requests for clarification:

1. There could be a lot more effort to engage with the acculturation literature. I fear that acculturation researchers will be left cold by the mix of intuitive findings and simplifications made in the models. In the discussion I would like to see specific recommendations deriving from the models: are the insights from the models consistent with specific empirical patterns? What do the models suggest are important factors that are not currently considered in acculturation research, but which could be measured? In other words, how exactly can these models guide future empirical work?

We thank the reviewer for this suggestion. We have added two paragraphs to the Discussion section (lines 395-428) in which we more extensively discuss the scope and limitations of our model, and try to situate the relevance of our model more clearly in light of the existing acculturation literature, especially with regard individual differences and changes/mismatches in acculturation attitudes. We also provide a suggestion for empirical research that can help improving our model with the ultimate aim of better understanding how acculturation attitudes affect the distribution of cultural traits in a population over time.

2. The term 'multicultural' needs to be clearly defined. In the present models it seems to denote an internal equilibrium where some individuals have one trait and other individuals have the other trait. I doubt whether this would count as 'multiculturalism' in either the acculturation literature or to lay people. There is no blending of traits, or single individuals holding multiple traits of different cultural origin. Using multicultural in this restricted sense with no definition will lead to confusion.

We added our definition of "a multicultural society" within the context of this study to the abstract (lines 29-30).

3. Please give examples, ideally drawing on the acculturation literature, of real-life traits subject to acculturation that might plausibly be (i) payoff-neutral, as in the first model, (ii) payoff-relevant, as in the second model, and (iii) subject to frequency effects, as in the third model. I felt the paper became very abstract at this point, and could have done with greater connection to the real world.

We thank the reviewer for this suggestion – examples can certainly help here to make the paper more readable. We added examples throughout the final paragraph of the Introduction (lines 145-161).

4. The overview of cultural evolution theory in the introduction is good, but I would cite and discuss Cavalli-Sforza & Feldman (1981) who already modelled the consequences of migration for cultural evolution (albeit not within an acculturation context).

We agree with the reviewer that at least a brief mention of the literature that has discussed migration in a cultural evolution framework was warranted in the Introduction section. We had already cited Cavalli-Sforza & Feldman (1981), but now also cite it as a reference for studies of migration in models of cultural evolution. We have also added references to three other more recent papers that focus on the role of migration within a cultural evolution framework (line 115).

5. Line 83: “Given the dynamic and complex nature of the acculturation process, it is likely that verbal reasoning alone is insufficient to fully appreciate the societal implications of migration on cultural change”. I think there could be more elaboration of this point for an acculturation audience who fail to see the value of modelling (see also point 1 above). Paul Smaldino has written some excellent articles recently explaining the benefits of models to psychologists.

We have now elaborated on the benefits of developing formal models for studying systems that are complex and change over time, and directed the readers to a recent book chapter by Paul Smaldino on this topic (lines 96-102).

6. Line 298: “More strikingly, ...[i]f immigrants and residents do not tend to interact much, the immigrants will not adopt the resident culture” - this seems rather obvious, and does not warrant the superlative “striking”

We softened the wording here, replacing “More strikingly” with “Moreover” (line 338).

7. Line 351: “these models” - citations needed

We rephrased this sentence (lines 389-391).

8. Please include all model code for the simulations, and ideally scripts for the analytical model analyses, as supplementary information, for others to check and build upon.

We have added a data availability statement that includes these (line 497).

Reviewer #4

This is a theoretical model examining the conditions under which we might expect to see the emergence of multicultural societies. This brings together perspectives from cultural psychology and theoretical cultural evolution.

Both the question asked and the perspectives taken are interesting and it is a very well written paper in my opinion. However, there appear to be some important assumptions underlying the results that have not been clearly specified or justified (see below). These will need to be addressed before publication.

Major points.

(1) It is unclear why learning events and migration events should be mutually exclusive. The text does not offer an explanation – is there a reason? There is a probability m of immigration and a probability $(1-m)$ of a learning interaction (subject to other parameters), so when m is high, learning is less likely. This seems like it would artificially strengthen the effects of increased migration – not only is there more of the I trait to start with, but when m is high, it is less likely to be replaced through interaction and learning.

There is another problem with this assumption. Usually, one would assume that as migration increases, the probability of the resident population interacting with a migrant also increases (this is just a sensible by-product of a well-mixed population). In this case this effect is watered down and that assumption will need to be justified and the consequences made clear at the outset of the paper.

We recognize that a clarification was in order here. Because our model is event-based and there are two possible events that can happen (immigration and interaction), the parameter m should be interpreted as denoting how much immigration there is relative to interaction (so it is not a count of migration events per unit time). Changing the value of m in the range between 0 and 1 allows for every possible ratio between the two (*i.e.* it allows $m/(1-m)$ to take any positive value), so this set-up is not restrictive about the relationship between the probabilities with which interaction and immigration events occur. Our model does not make any explicit assumptions about the time that passes between events, so a higher value of m therefore does not have to mean a higher number of migration events per unit time (or a

lower number of interaction events per unit time), but rather just a higher rate of migration relative to the rate of interaction. Our model can therefore not distinguish between situations where the total number of events per unit time is high or low, as in the example that the reviewer gives. If the number of immigration and interaction events per unit time would both go up, m would stay (approximately) equal. We have added a clarification in the “The model” section that briefly explains this rationale (lines 180-182).

(2) The model assumes that immigration is constant. The I trait is constantly flowing into the population and can also be transmitted through learning. However, the resident R trait can only spread through learning because the resident population cannot reproduce and replace immigrants. When interaction is low, this heavily favours the immigrant trait.

This assumption explains the finding that high levels of interaction favour the persistence of both traits and that low levels lead to fixation of the immigrant trait. If both populations could reproduce and replace one another, one might expect the opposite to be true (i.e. that low levels of interaction would maintain pockets of both traits and that high levels of interaction would favour a higher fitness trait, for example.)

The issue with this is shown clearly in Figure 3 where zero interaction on the left of the x-axis leads to fixation of the immigrant trait. It is also exemplified by Lines:

230-231 : ‘...In other words, even if the resident cultural trait is superior, it will be lost from the population if immigrants and residents rarely interact with each other’ and

311-313 : ‘We predict that constant payoff differences (resident advantage or immigrant advantage) are only of importance if individuals of both cultures interact with a frequency that is sufficiently high (and even then, they are far from all-determining)’, which are both highly counter-intuitive statements. With a strong fitness advantage, no trait should die out, all else being equal.

We agree with the reviewer that our assumption of a constant immigration pressure is important for the model outcomes. Although it is necessary to make simplifying assumptions such as this one when constructing a model, we recognize the need to address the importance of this assumption for the outcomes explicitly. We added a paragraph in the Discussion section addressing issues related to a fixed population size, constant immigration and related assumptions (lines 408-428).

(3) It is not clear why the simulations (e.g. in Figure 2) so frequently show the I trait dying out when (a) migration should be constant and therefore a lower bound on the frequency of I might reasonably be expected to be m and (b) the analytical model shows two equilibria, neither of which is extinction of the I trait? I’m sure there is a reason but could the authors add some text to the discussion of Figure 2 to address this?

The reviewer correctly mentions that for any positive migration rate, the type I trait can never go fully extinct. In specific simulations for low m it can still be that there are no I individuals in the population at any given point in time (due to stochastic effects), but fixation of trait R is not an equilibrium. We added a brief explanation in the caption of Figure 2 to reflect this (lines 247-251); see also the third Minor Point of this reviewer.

Minor Points

Figure 1. This is interesting and helpful for those not familiar with this theory. It would also be very helpful for the authors to clearly show what each of the four endpoints would mean in their model. It looks like some of them are not possible in the model and it is important to know that not all outcomes are possible. For example, multiculturalism in the model means that both traits survive in some proportion. Both traits cannot exist in one individual so perhaps ‘assimilation’ is not possible but ‘separation’ is possible? This should be made clear in the figure.

We added a clarification of what the four endpoints would represent in the figure legend (we have not formulated this too much in model terms yet because the model still needs to be explained at this point in the text). In principle, all these outcomes are possible in our model,

when interpreting them strictly in terms of the two dimensions that we consider (the degree of interaction between the host and immigrant culture, or 'social integration' of immigrants, and the degree to which the immigrants maintain their cultural identity).

Equation 1. In general, I usually maintain that equations should not appear in the text if not fully explained. Perhaps either point to the methods and omit the equation at this point or explain its terms fully. It does not fulfill a useful purpose at that point in the text.

We thank the reviewer for this valuable suggestion. Although we are restricted to keeping a detailed Methods section at the end of the text because of the journal format, we have now added a more detailed explanation of this main equation where it appears in the text (lines 192-204).

Figure 2. When m is zero, there is no immigration. If the population is initially all of type R, all boxes in the left hand corner of the plots should be dark blue (a frequency of 0 for type I) – why are they showing a frequency of ~0.5?

The reviewer is correct that a migration rate of 0 should lead to fixation of type R, and all boxes should be blue if that were the case. However, as mentioned in our response to Major Point 3 of this reviewer, we have only considered values of m above zero, which explains the non-zero frequencies of type I for the lowest migration rate we considered (0.001). We thank the reviewer for spotting this – this was not entirely clear in the figure caption of the original manuscript. We now explicitly mention the range of m we considered in the caption of Figure 2 (line 248).

Reviewers' comments:

Reviewer #2 (Remarks to the Author):

This revision presents basically the same models and results, but with material added to the Introduction and Discussion to aid comprehension. It achieves this, and is almost publishable. I still think it makes a valuable and novel contribution to the literature. I also commend the availability of the model on github. I have a few minor comments remaining about presentation of the model.

1. I realise the journal imposes its sub-section ordering, but one cannot understand the equations without reference to the parameter definitions, which are not explained until Table 1 in the Methods. For example, delta-c is not defined in the text but discussed nevertheless on page 10. Move Table 1 to the main text, so that the parameters are known.
2. Line 256: add some detail about how payoffs were implemented, otherwise again the reader has to refer down to the methods in order for it to make any sense. E.g. say that switching probability delta-S is now determined by both conservatism values and payoff differences.
3. Clarify in Table 1 the direction of delta terms, e.g. does $\text{delta-c} = c_i - c_r$ or $c_r - c_i$?
4. Line 440: "For simplicity, we assume that acculturation orientations are linked to cultural traits, so if individuals change their cultural trait, they also change their acculturation orientation" - this stood out to me second time reading, as an important assumption that is not highlighted enough. In essence, even the first neutral model isn't really neutral: it involves the transmission not just of arbitrary markers but also acculturation strategies. In particular, copying a migrant also entails becoming like a migrant in conservativeness. What happens if this assumption is changed and acculturation attitudes are not transmitted? At least this should be made more explicit and not hidden in the Methods.

Reviewer #4 (Remarks to the Author):

The various points of clarification have been addressed well by the authors and I thank them for their time and effort in responding to my review. I do continue to have some reservations about the model assumptions, some of which have a direct bearing on some quite counter-intuitive results.

Response to point 1.

While I appreciate the response to this point, I am worried that the authors have not directly tackled the core issue. The problem, as I see it, is that there are two possible events in the model that are mutually exclusive – immigration or interaction. I can see no reason that these should be mutually exclusive even in a model where, of course, sometimes-awkward simplifications are necessary.

It seems that the addition of a step in the model that allows interaction to occur independently of the rate of migration would be relatively straightforward in the simulations (if not in the analytical model). This would be a useful check on the extent to which the assumption is driving the results and could perhaps form the basis of a short appendix, for the sake of completeness. I anticipate that the effects would be large but, of course, they may not be (models are wonderful that way!)

The authors state that they are examining the rate of migration relative to the rate of interaction. The language throughout the results and discussion imply that the rate of migration is being examined directly, rather than relatively and this should also be addressed (e.g line 233, caption of figure 2 etc.) especially if the authors feel that additional simulations are impossible.

In other words, I believe the inter-dependence of migration rate and interaction rate should be very clearly stated throughout the manuscript, and/or examined directly through further simulation to precisely establish its effects on the results.

Response to point 2.

Again, I thank the authors for their response. I am concerned that the section added to the text does not deal with the worry that the counter-intuitive results of the model (i.e. that high levels of interaction favour the persistence of both traits and that low levels lead to fixation of the immigrant trait) are to some extent artefacts of the assumption that the immigrant trait has two 'modes of spread', if you will, and the resident trait has just one, artificially favouring the immigrant trait when interaction is low. This possibility needs to be addressed much more directly in my opinion.

‘The effect of acculturation on the evolution of a multicultural society’
Manuscript by: Erten, Van den Berg and Weissing

Response to reviewers (second round)

This document contains a point-by-point description of how we revised our manuscript in response to the reviewers’ concerns. We again thank the reviewers for their time and effort in giving detailed constructive criticism on our manuscript. Reviewer remarks will be printed in black, and a concise description of our response will be printed in blue.

Reviewer #2

This revision presents basically the same models and results, but with material added to the Introduction and Discussion to aid comprehension. It achieves this, and is almost publishable. I still think it makes a valuable and novel contribution to the literature. I also commend the availability of the model on github. I have a few minor comments remaining about presentation of the model.

1. I realise the journal imposes its sub-section ordering, but one cannot understand the equations without reference to the parameter definitions, which are not explained until Table 1 in the Methods. For example, delta-c is not defined in the text but discussed nevertheless on page 10. Move Table 1 to the main text, so that the parameters are known.

We moved Table 1 to the main text.

2. Line 256: add some detail about how payoffs were implemented, otherwise again the reader has to refer down to the methods in order for it to make any sense. E.g. say that switching probability delta-S is now determined by both conservatism values and payoff differences.

This is a good point – more clarification was in order here. We added two sentences to this effect (lines 264-268).

3. Clarify in Table 1 the direction of delta terms, e.g. does $\delta\text{-c} = c_i - c_r$ or $c_r - c_i$?

We have added this clarification in the table.

4. Line 440: “For simplicity, we assume that acculturation orientations are linked to cultural traits, so if individuals change their cultural trait, they also change their acculturation orientation” - this stood out to me second time reading, as an important assumption that is not highlighted enough. In essence, even the first neutral model isn’t really neutral: it involves the transmission not just of arbitrary markers but also acculturation strategies. In particular, copying a migrant also entails becoming like a migrant in conservativeness. What happens if this assumption is changed and acculturation attitudes are not transmitted? At least this should be made more explicit and not hidden in the Methods.

We have moved this sentence from the Methods section to the ‘The Model’ section, so that the reader explicitly encounters this assumption early in the text (lines 208-211) and is therefore made more aware of this element of the model set-up. We made this assumption to keep a clear focus on between-culture differences. Therefore, we did not allow any individual differences within cultures, neither in cultural trait nor in acculturation orientation. However, we do think that relaxing this assumption would be one of the most interesting extensions of our model. We had already dedicated a paragraph to this in the discussion section, but have now added a sentence to this paragraph in which we highlight this assumption more explicitly as part of this approach (lines 412-415).

Reviewer #4

The various points of clarification have been addressed well by the authors and I thank them for their time and effort in responding to my review. I do continue to have some reservations about the model assumptions, some of which have a direct bearing on some quite counter-intuitive results.

Response to point 1.

While I appreciate the response to this point, I am worried that the authors have not directly tackled the core issue. The problem, as I see it, is that there are two possible events in the model that are mutually exclusive – immigration or interaction. I can see no reason that these should be mutually exclusive even in a model where, of course, sometimes-awkward simplifications are necessary.

It seems that the addition of a step in the model that allows interaction to occur independently of the rate of migration would be relatively straightforward in the simulations (if not in the analytical model). This would be a useful check on the extent to which the assumption is driving the results and could perhaps form the basis of a short appendix, for the sake of completeness. I anticipate that the effects would be large but, of course, they may not be (models are wonderful that way!)

We thank the reviewer for this valuable suggestion. We have now developed a generalized version of the simulation model in which interaction events and immigration events occur with separate (independent) probabilities in each time step. This means that rather than having either an immigration event or an interaction event in each time step, we can now have neither, either, or both types of events happening in each time step. We have taken up the reviewer's suggestion to write a short appendix in which we present this model, and compare its results with the original model presented in the main text. As is visible from this comparison, if the ratio of the probability of an interaction event to the probability of a migration events is the same, both models produce virtually the same outcome. We now direct the reader's attention to this model in the 'The Model' section (lines 182-184). The code of the generalized model will also be made available on GitHub, where the code of the original simulation model can also be found.

The authors state that they are examining the rate of migration relative to the rate of interaction. The language throughout the results and discussion imply that the rate of migration is being examined directly, rather than relatively and this should also be addressed (e.g line 233, caption of figure 2 etc.) especially if the authors feel that additional simulations are impossible.

We have added some more explicit clarifications about the migration rate throughout the text, perhaps most importantly in Table 1, which is now also placed earlier in the text (in the 'The Model' section rather than in the 'Methods' section). We also added clarifications at the start of the results section (lines 228-229) and at the point where we introduce M (lines 306-307).

In other words, I believe the inter-dependence of migration rate and interaction rate should be very clearly stated throughout the manuscript, and/or examined directly through further simulation to precisely establish its effects on the results.

As is now clear from the newly added Supplementary Information, the fact that the probabilities of migration and interaction are determined by the same parameter does not affect model outcomes – a model where both are determined by separate parameters produces virtually the same results as long as the ratio of both probabilities is the same. Nevertheless, we agree with the reviewer that it is good to be more explicit about the exact meaning of the parameter m throughout the text – this can only enhance the general clarity

of the way our model is presented. We have therefore been more explicit about this throughout the text, including in Table 1, which serves as a reference list of all parameters and is now placed earlier in the text.

Response to point 2.

Again, I thank the authors for their response. I am concerned that the section added to the text does not deal with the worry that the counter-intuitive results of the model (i.e. that high levels of interaction favour the persistence of both traits and that low levels lead to fixation of the immigrant trait) are to some extent artefacts of the assumption that the immigrant trait has two 'modes of spread', if you will, and the resident trait has just one, artificially favouring the immigrant trait when interaction is low. This possibility needs to be addressed much more directly in my opinion.

We have now dedicated a full paragraph to this issue, in which we more explicitly address the concern that our assumption of a constant migration rate is an important determinant of our model outcomes. We now also speculate on the possible ramifications of relaxing this assumption in this paragraph (lines 425-441).

Reviewers' comments:

Reviewer #4 (Remarks to the Author):

I thank the authors for extending and generalizing their model, which I appreciate was a frustrating request. It is great to see that the model results are upheld under certain circumstances.

However, the response from the authors says 'the fact that the probabilities of migration and interaction are determined by the same parameter does not affect model outcomes'. The concern was not the use of a single parameter, of course, but that the rate of migration and interaction were interdependent. Because the authors have chosen to examine their new model only when the ratio of ρ_i to ρ_m is equal to $(1-m)/m$, we still do not have a complete picture although that should now be very easy to achieve and discuss briefly.

It is important to ask - what happens when the ratio of ρ_i to ρ_m is not equal to $(1-m)/m$?

This is the equivalent of asking what happens when the rate of migration and interaction are not dependent on one another.

I thank the authors for their response to the rest of the comments, which I think have clarified the potential impact of some important assumptions.

‘The effect of acculturation on the evolution of a multicultural society’
Manuscript by: Erten, Van den Berg and Weissing

Response to reviewers (third round)

This document contains a description of how we revised our manuscript in response to the concerns of Reviewer #4. Since the comments of the reviewer only pertained to the Supplementary Information, all changes we have made are there – the main text has remained unchanged in this revision. We again thank the reviewer for their constructive criticisms on this manuscript, which have again led to some real improvements (see below). Our response to the reviewer’s comments is found below, including a description of the changes we have made to the SI. Reviewer remarks will be printed in black, and a concise description of our response will be printed in blue.

Reviewer #4

I thank the authors for extending and generalizing their model, which I appreciate was a frustrating request. It is great to see that the model results are upheld under certain circumstances.

However, the response from the authors says ‘the fact that the probabilities of migration and interaction are determined by the same parameter does not affect model outcomes’. The concern was not the use of a single parameter, of course, but that the rate of migration and interaction were interdependent. Because the authors have chosen to examine their new model only when the ratio of ρ_i to ρ_m is equal to $(1-m)/m$, we still do not have a complete picture although that should now be very easy to achieve and discuss briefly.

It is important to ask - what happens when the ratio of ρ_i to ρ_m is not equal to $(1-m)/m$?

This is the equivalent of asking what happens when the rate of migration and interaction are not dependent on one another.

I thank the authors for their response to the rest of the comments, which I think have clarified the potential impact of some important assumptions.

The reviewer requests that we run simulations for which ρ_i/ρ_m is not equal to $(1-m)/m$. Based on this response, we suspect that our use of terminology may have been somewhat confusing. To be clear, the parameters ρ_i and ρ_m of the generalized model have completely replaced parameter m of the original model. Hence, m is no longer part of the generalized model. The values of ρ_i and ρ_m were not constricted for the simulations that we presented for the generalized model: they did not have to sum up to one, and their ratio was not constrained by the parameter m . The only thing we did is **compare** the cases of the original model and the generalized model where $(1-m)/m$ in the former was equal to ρ_i/ρ_m in the latter, by placing the subgraphs for these outcomes in the same column in Figure S1. Hence, when one wants to compare cases where ρ_i/ρ_m is **not** equal to $(1-m)/m$ (as the reviewer suggests), all one has to do is compare outcomes of the original model with outcomes of the generalized model that are not in the same column.

In view of the reviewer’s comments, we realize that the introduction of two new parameters may have been confusing. We therefore changed our notation, using the symbol m for the rate of migration (previously ρ_m) and the symbol i for the rate of interaction (previously ρ_i). This way, it is clearer that m controls the rate of migration in both the original and the generalized model (although in the original model, it also determines the rate of interaction, whereas it does not in the generalized model). In addition to changing the

symbols for these parameters, we have also completely rewritten the text of the SI to more clearly explain the function of both parameters in the generalized model, and their relationship to m in the original model.

Additionally, we have met the reviewer's request to show a more comprehensive overview of outcomes of the generalized model (including outcomes where ρ_i/ρ_m [now i/m] is not equal to $(1-m)/m$ for any of the cases of the original model that are shown in Figure S1), and now also consider a wider range of immigration rates. For easy comparison between both models, we have now highlighted the cases of the generalized model (in yellow) for which i/m is equal to $(1-m)/m$ for the original model subgraph in the same column. Apart from comparing outcomes between both models, showing more parameter combinations for the generalized model now allows us to assess the effect of the migration rate m and the interaction rate i within the generalized model. Interestingly, this comparison of results within the generalized model reveals that the migration rate has a much stronger effect on the outcome than the interaction rate. This is a conclusion we would not have arrived at if the reviewer had not asked us to show more results, so we thank the reviewer for this input.

REVIEWERS' COMMENTS:

Reviewer #4 (Remarks to the Author):

I would like to thank the authors for their response to the review and their willingness to expand on and clarify the results of the generalized model.

I believe my concerns have been addressed.